# Phylogenetic Studies and Metabolite Analysis of *Sticta* Species from Colombia and Chile by Ultra-High Performance Liquid Chromatography-High Resolution-Q-Orbitrap-Mass Spectrometry

**DOI:** 10.3390/metabo12020156

**Published:** 2022-02-08

**Authors:** Laura Albornoz, Alfredo Torres-Benítez, Miguel Moreno-Palacios, Mario J. Simirgiotis, Saúl A. Montoya-Serrano, Beatriz Sepulveda, Elena Stashenko, Olimpo García-Beltrán, Carlos Areche

**Affiliations:** 1Departamento de Química, Facultad de Ciencias, Universidad de Chile, Las Palmeras 3425, Nuñoa, Santiago 7800024, Chile; albornoznocachomail.c@hotmail.com; 2Instituto de Farmacia, Facultad de Ciencias, Universidad Austral de Chile, Campus Isla Teja, Valdivia 5090000, Chile; aljotobe19@hotmail.com (A.T.-B.); mario.simirgiotis@uach.cl (M.J.S.); 3Laboratorio de Biología Evolutiva de Vertebrados, Departamento de Ciencias Biológicas, Universidad de Los Andes, Bogotá 111711, Colombia; mc.morenop@uniandes.edu.co; 4Laboratorio de Toxicología, Seccional Tolima, Instituto Nacional de Medicina Legal y Ciencias Forenses, Ibagué 730006, Colombia; samontoya@medicinalegal.gov.co; 5Departamento de Ciencias Químicas, Universidad Andres Bello, Campus Viña del Mar, Quillota 980, Viña del Mar 2520000, Chile; bsepulveda@uc.cl; 6Research Center of Excellence CENIVAM, CIBIMOL, Universidad Industrial de Santander, Building 45, UIS, Carrera 27, Calle 9, Bucaramanga 680002, Colombia; elena@tucan.uis.edu.co; 7Centro Integrativo de Biología y Química Aplicada (CIBQA), Universidad Bernardo O’Higgins, General Gana 1702, Santiago 8370854, Chile; 8Facultad de Ciencias Naturales y Matemáticas, Universidad de Ibagué, Carrera 22 Calle 67, Ibagué 730002, Colombia

**Keywords:** chemotaxonomyc, lichens, metabolomics, *Sticta*, phylogenetic, UHPLC-MS-MS

## Abstract

Eleven species of lichens of the genus *Sticta*, ten of which were collected in Colombia (*S. pseudosylvatica S. luteocyphellata S.* cf. *andina S.* cf. *hypoglabra, S. cordillerana, S.* cf. *gyalocarpa S. leucoblepharis, S. parahumboldtii S. impressula, S. ocaniensis*) and one collected in Chile (*S. lineariloba*), were analyzed for the first time using hyphenated liquid chromatography with high-resolution mass spectrometry. In the metabolomic analysis, a total of 189 peaks were tentatively detected; the analyses were divided in five (5) groups of compounds comprising lipids, small phenolic compounds, saturated acids, terpenes, and typical phenolic lichen compounds such as depsides, depsidones and anthraquinones. The metabolome profiles of these eleven species are important since some compounds were identified as chemical markers for the fast identification of *Sticta* lichens for the first time. Finally, the usefulness of chemical compounds in comparison to traditional morphological traits to the study of ancestor–descendant relationships in the genus was assessed. Chemical and morphological consensus trees were not consistent with each other and recovered different relationships between taxa.

## 1. Introduction

Lichens constitute a mutualistic symbiosis with green algae and/or cyanobacteria [1], and in some cases present a tripartite symbiosis between different mycobionts and photobionts known as photosymbiodemas [2,3,4]. The genus *Sticta* (Schreb.) Ach. is the most species-diverse group of macrolichens in the family Lobariaceae, with about 120 species, and is characterized by a heteromeric thallus, with wide and rounded or elongated and truncated lobes, sometimes overlapping and rarely polyphilic, with smooth upper surface or with light ribs that may carry isidia or soralia, and the presence of cyphelas on the ventral surface of variable sizes dispersed in the tomentum [5,6,7,8,9,10,11]. In South America, the genus is found in Andean, sub-Andean, and moorland ecosystems, and develops in substrates of bark, soil, wood and rocks [5,6,7,8,9,10,11,12].

Chemical studies of the genus *Sticta* are scarce. In *Sticta* and other genera, most compounds are of fungal origin and their chemical variety is related to the environmental conditions in which the species develops [13,14], which is observed in this work. From the earliest reports found for this genus, triterpenes were identified in the species *S. billardierii*, *S. coronata*, *S. colensoi*, and *S. favicans* [15,16]. The latest reports on species such as *S. fuliginosa*, *S. weigelia*, *S. caulescens* and *S. santessonii* show the presence of substances such as 7β-acetoxy-22-hydroxy hopane usnic acid, ursolic acid, ergosterol peroxide and β-sitosterol [17]. The *S. nylanderiana* ethyl-3-formyl-2,4-dihydroxy-5,6-dimethylbenzoate, methyl-2,4-dihydroxy-3,5,6-trimethylbenzoate, orsellinic acid, methyl orsellinate, ethyl orsellinate, lecanoric acid, 4-O-methyl gyrophoric acid and retigeric acid B compounds were isolated and identified [18]. In an unidentified species of the genus *Sticta*, the compounds stictamides A-C were isolated and identified. In addition, stictamide A acts as an inhibitor of the MMP12 protease (metallopeptidase 12) [19].

Currently, for the identification and elucidation of metabolites in complex extracts, the technique of ultra-high performance liquid chromatography-diode array detection (UHPLC-PDA) coupled to an electrospray ionization tandem mass spectrometer (ESI–MS–MS) [20,21,22,23,24,25,26] is used. The Q-Exactive Focus is a hybrid high-resolution mass spectrometer used to detect and quantify small organic compounds via high-resolution accurate mass spectrometry. This machine combines UHPLC-PDA (ultrahigh pressure liquid chromatography with photodiode array) with an orbitrap, a quadrupole (Q) and a high-resolution collision cell (HRCD), which allows for high resolution diagnostic untargeted metabolomics and accurate determination of fragments [15,16,17,18,19,20,21,22,23,24,25,26,27,28]. This technique has been useful for the chemical study of several lichens of the genera *Ramalina*, *Parmotrema* and *Usnea* [23,24,25,26,27,28,29,30] conducted by our research group and others. 

In this research, we analyzed the phytochemical profile of several species of the *Sticta* genus collected in different geographical regions of Colombia and Chile, based on UHPLC-DAD coupled with high-resolution electrospray ionization tandem mass spectrometry (ESI-MS-MS) for the first time. The eleven (11) lichens of the genus *Sticta* analyzed were *S. pseudosylvatica S. luteocyphellata S.* cf. *andina S.* cf. *hypoglabra*, *S. cordillerana*, *S.* cf. *gyalocarpa S. leucoblepharis*, *S. parahumboldtii S. impressula*, *S. ocaniensis* and *S. Lineariloba* (Figure 1). Based on the quantity and variety of chemical compounds found in the studied specimens, we complemented the chemical characterization with a maximum parsimony analysis based on the hypothesis that phytochemical compounds in lichens may show a phylogenetic signal, thus proving utility for chemotaxonomy.

## 2. Results and Discussion

In this study, the metabolome profile of eleven species of *Sticta* lichens are reported for the first time. Lichen substances have gained considerable attention due to their potential health benefits and possible food nutraceutical or biotechnological applications [31]. Such compounds consist mostly of aliphatic and aromatic substances which proved to have biological and pharmacological activities compared to higher plants, in particular several depsidones and depsides which proved to be antioxidant, cytotoxic and anti-inflammatory agents, among several other bioactivities reported from the genus [31,32,33,34,35]. Their identification by hyphenated mass spectrometry liquid chromatography techniques is highly important for the fast metabolite profiling of these important biodiverse organisms, and the possibility of finding biomarkers that could be of help for their identification.

### 2.1. Identification of Metabolites in 11 Lichen Species

In this study, eleven species of the genus *Sticta* were analyzed and Figure 2 shows the chromatograms for the species *S. pseudosylvatica* and *S. lineariloba*. In the metabolomic analysis, a total of 189 peaks were detected, 41 of which were unknown compounds, and the analyses were divided in five groups of compounds as explained below (Table 1).

#### 2.1.1. Saturated Organic Acids

Peak 1 was tentatively identified as gluconic acid (C_6_H_11_O_7_), peak 3 as manitol, peak 4 as arabic acid, peak 5 as citric acid and peak 10 as its isomer isocitric acid (C_6_H_7_O_7_), peak 7 as 4-ethyl-2-ethylisophthalic acid (C_10_H_9_O_4_) with peak 2 as its isomer 2-ethylisophthalic acid, peak 15 as the derivative 2-hydroxyisophthalic acid, peak 17 was identified as 4-O-demethylglomellic acid (C_24_H_25_O_9_), peak 30 as 1,5-pentanedicarboxylic acid (C_7_H_11_O_4_) and peak 44 as 2,4-dicarboxy-3-hydroxy-5-methoxytoluene (C_10_H_9_O_6_)

#### 2.1.2. Small Phenolic Compounds

Peaks 14 and 16 were identified as trihydroxy benzalaldehyde (C_7_H_5_O_4_) and 2,4-dihydroxy benzaldehyde (C_7_H_5_O_3_), peak 25 as 5,7-dihydroxy-6-methylphthalide (C_9_H_7_O_4_), peak 39 as metil-2,6-dihidroxibenzoate (C_8_H_7_O_4_), peak 51 as 2,4-dihydroxy benzaldehyde (C_7_H_5_O_3_), peak 53 as 4-ethoxy-3-formyl-2-hydroxy-6-methylbenzoic acid (C_11_H_11_O_5_), peak 95 5,7-dihydroxy-6-methylphthalide derivative (C_9_H_7_O_3_), peak 132 as ethyl 2,4-dihydroxy-6-n-nonylbenzoate (C_18_H_27_O_4_).

#### 2.1.3. Typical Lichenic Phenolic Compounds (Depsides, Depsidones and Anthraquinones)

Several compounds were identified as the typical types of compounds occurring in lichens in our *Sticta* species. Peak 20, with a deprotonated molecule at *m/z*: 413.1569, was identified as grayanic acid (C_23_H_25_O_7_), peak 23 was identified as atranol (C_8_H_7_O_3_) [25], peak 32 as didechlorolecideoidin (C_17_ H_13_O_7_) [36] showing diagnostic fragments at *m/z*: 209.0456; 285.0776; 151.0396; 179.0347 and 123.0443, peak 34 was identified as orsellinic acid (C_8_H_7_O_4_), peak 37 as nor 8′-methylconstictic acid (C_21_H_19_O_11_), showing a diagnostic daughter fragment at *m/z*: 209.0455, peak 40 as hypostictic acid isomer (C_19_H_15_O_8_), peak 42 as fumarprotocetraric acid derivative (C_17_H_11_O_6_), peak 52 as consalizinic acid derivative I (C_19_H_13_O_11,_ with diagnostic daughter ions at m/z: 373.0573; 387.0373; 225.0406 and 177.0193, and peak 55 as consalizinic acid derivative II, with ions at *m/z*: 401.0524; 417.0474 and 373.0574), peak 56 and 57 as an cynodontin, citreorosein isomer or consalizinic acid derivative I isomer, respectively, peak 59 as 1,4,5,6,8-pentahydroxy-3-ethylanthraquinone (C_15_H_9_O_7_), peak 62, with an ion at *m/z*: 403.0681 was identified as a haemathamnolic acid isomer (C_19_H_15_O_10_) and peak 64 as constictic acid (C_19_H_13_O_10_), peak 63 as a fumarprotocetraric acid derivative (C_17_H_11_O_6_), peak 65 as hypostictic acid isomer (C_19_H_15_O_8_), producing daughter ions at *m/z*: 327.0883, 195.0664 and 179.0347 peak 66 as terphenylquinones thelephoric acid (C_18_H_7_O_8_) and peak 67 as methylstictic acid (C_20_H_15_O_9_, ions at 371.0779 and 193.0504). Peak 68 was tentatively identified as 8′-metilconstictic acid isomer (C_21_H_19_O_11_), and peak 69 as protocetraric acid (C_18_H_13_O_9_) [29], peak 70 as hypoconstictic acid (C_19_H_15_O_9_), peak 77 as menegazziaic acid (C_18_H_13_O_9_) [37] with ionic fragments at *m/z*: 311.0570; 255.0666; 329.0679, peak 78 as norstictic acid (C_18_H_11_O_9_ ions at *m/z*: 327.0526; 151.0396 and 123.0444), peak 80 as the antioxidant agent physodalic acid [35] (C_20_H_15_O_10,_ MS^2^ peaks at *m/z*: 359.0417; 315.0520; 343.0832; 387.0367; 373.0573 and 401.0525), peak 82 as derivative methyl 8-hydroxy-4-O-demethylbarbatate (C_19_H_19_O_9_), peak 83 as 12,13,15-trihydroxy-9-octadecenoic acid (C_18_H_33_O_5_), peak 84 as the cytotoxic compound haemoventosin (C_15_H_11_O_7_) [38], with ions at 259.0619, 231.0667, 189.0560, peak 86 as conhypoprotocetraric acid or convirensic acid (C_18_H_15_O_8_) [29], peak 87 as 4-O-dimethylbaeomycesic acid (C_18_H_15_O_8,_ 181.0714; 163.0397 and 137.0236), a methyl derivative of baeomycesic acid [39], peak 88 as orsellinic acid isomer (C_8_H_7_O_4,_ 123.0440; 149.0235), peak 89 as lecanoric acid (C_16_H_13_O_7,_ 167.0345; 123.0443; 149.0238) [23], peak 90 as constictic acid isomer (C_19_H_13_O_10_), peak 92 with a deprotonated molecule at *m/z*: 235.0615 and daughter ion at *m/z*: 181.0504 was identified as 2-methyl-5-hydroxy-6-hydroxymethyl-7-methoxychromone (C_12_H_11_O_5_), peak 96 was identified as criptostictic acid derivative (C_18_H_11_O_8_), peak 105 as stictic acid (C_19_H_13_O_9_) [25], peak 106 as the typical lichen anthraquinone parietin (C_16_H_11_O_5_), while peak 108 was identified as evernic acid isomer (C_17_H_15_O_7_), peak 109 as hypoconstictic acid and peak 110 as cryptostictic acid [40] (C_19_H_15_O_9,_ diagnostic daughter ions at *m/z*: 267.0661; 343.0825, 311.05067 and 239.0710)_._ peak 113 as salazinic acid isomer (C_18_H_11_O_1_), peak 117 with a deprotonated molecule at *m/z*: 323.0556 was identified as pulvinic acid derivative I (C_18_H_11_O_6_), producing ions at *m/z*: 133.0286; 117.0335. Peak 119, with a parent ion at *m/z*: 345.0989, was identified as 4-O-demethylbarbatic acid (C_18_H_17_O_7_) [29], peak 121 as methyl orsellinate [37] and peak 123 as gyrophoric acid. Peak 125, producing fragments at *m/z*: 313.0723; 135.0444 and 179.0348, was identified as hyposalazinic acid (C_18_H_13_O_8_) [37], peak 127 as an isomer of orsellinic acid [23], peak 124 as galbinic acid (C_20_H_13_O_11_, 403.0681; 371.0417; 401.0524; 327.0518 and 149.0239) [37], peak 129 as norstictic acid (C_18_H_11_O_9,_ 27.0517; 227.0716; 151.0390; 243.0297) [1,2], peak 133 as evernic acid (C_17_H_15_O_7_) [25], peak 131 as loxodinol isomer (C_25_H_29_O_9_), peak 136 as the dibenzophenone strepsilin (C_15_H_9_O_5_), peak 146 as squamatic acid and peak 147 as the depsone picrolichenic acid (C_25_H_29_O_7_), peak 144 and 145 as hydroxytetracosapentaenoic acid derivative (C_24_H_37_O_3_) and hydroxytrioxotricosanoic acid (C_23_H_39_O_6_), respectively. Peak 150, with an ion at *m/z*: 457.2244, was identified as 2,2′-di-O-methylanziaic acid (C_26_H_33_O_7_) [41], peak 154 was determined to be pulvinic acid, (C_18_H_11_O_5_, ions at *m/z*: 117.0338; 263.0713), peak 164 as a pulvinic acid derivative (C_19_H_13_O_5_), peak 164 as a pulvinic acid derivative of 321.0770 (C_19_H_13_O_5_), and peak 165 as another isomer of pulvinic acid (C_18_H_11_O_5_), while peak 155 was identified as 4-O-demethylbarbatic acid (C_18_H_17_O_7_, diagnostic ions at *m/z*: 123.0443; 137.0237; 181.0502). Peak 156 was determined as soromic acid (ions at *m/z*: 313.0726; 181.0502; 179.0347; 327.0520; 269.0826 and 285.0776), peak 157 as methylgyrophoric acid (C_25_H_21_O_10,_ diagnostic ions at *m/z*: 149.0238; 123.0442; 167.0346 and 317.0671), and peak 158 as evernic acid isomer. Similarly, peak 159 was identified as anthraquinones skyrin (C_30_H_17_O_10_), peak 160 as angardianic acid (C_19_H_35_O_4_) [42], peak 166, with an ion at *m/z*: 551.1197, was identified as furfuric acid isomer (C_28_H_23_O_12_, producing fragments at *m/z*: 371.0784; 193.0504; 179.0347; 207.0297 and 193.0504), peak 172 as barbatic acid (C_19_H_19_O_7_) [25], peak 174 as thamnolic acid, peak 175 as orsenillic acid derivative II (C_8_H_7_O_4_), peak 177 as lobaric acid (C_25_H_27_O_8_ with ions at *m/z*: 411.1824; 367.1811), peak 180 as hypothamnolic acid (C_19_H_17_O_10_ with ions at *m/z*: 209.0456; 181.0499) [43], peak 184 as usnic acid (C_18_H_15_O_7_) [25], peak 185 as either nephromopsic acid or roccellaric acid (C_19_H_33_O_4_) [44], and finally, peak 187 was identified as the cytotoxic compound perlatolic acid [45], peak 188 as the antibacterial compound caperatic acid [46] and peak 189 as atranorin.

#### 2.1.4. Terpenes

Peak 112 was identified as retigeric acid B (C_30_H_45_O_6_), while peak 111 with an ion at *m/z*: 515.3025 was tentatively identified as a retigeric acid derivative (C_30_H_43_O_7_).

#### 2.1.5. Lipids 

Oxylipins polyunsaturated fatty acids are an important dietary compounds, and can be found in edible fruits by HPLC orbitrap mass spectrometry [47] and also can be found in useful plants [48] and lichens [49]. In this study, several fatty acids including saturated fats and oxylipins were found using this technique in *Sticta* lichens. Peak 33, with a parent ion at *m/z*: 555.3047, was identified as decahydroxyoxopentacosanoic acid (C_25_H_47_O_13_); peak 43, with a parent ion at *m/z*: 187.0977, was determined to be 4,5-dihydroxy-2-nonenoic acid (C_9_H_15_O_4_); peak 72 as 12,13,15-trihydroxy-9-octadecenoic acid (C_18_H_15_O_5_), while peaks 91 and 94 were determined as pentahydroxytetracosanoic acid (C_24_H_47_O_7_) and heptahydroxytrioxooctadecanoic acid (C_18_H_29_O_12_), respectively. In the same manner, peaks 102–104 were identified as heptahydroxytetraoxoicosanoic acid (C_20_H_31_O_13_), tetrahydroxytricosanoic acid (C_23_H_45_O_6_), and tetrahydroxytrioxoundecanoic acid (C_11_H_15_O_9_), respectively. Peak 115 was assigned as 9,10-dihydroxyoctadecatrienoic acid (C_18_H_29_O_4_) and peak 118 as 9,10,12 trihydroxytriacontaheptaenoic acid; peak 122 as eptahydroxyetraoxoicosanoic acid (C_20_H_31_O_13_); peak 126 as hydroxytetracosapentaenoic acid (C_24_H_37_O_3_); and peak 128 as dihydroxyoctadecenoic acid (C_18_H_33_O_4_). Peak 130 was tentatively identified as dihydroxyoctadec-6-enoic acid (C_18_H_33_O_4_); peak 134 as a protocetraric acid isomer (C_18_H_13_O_9_) [29]; peak 139 as hexahydroxytrioxooctacosatrienoic acid (C_28_H_43_O_11_); peak 140 as nonahydroxyoctacosatetraenoic acid (C_28_H_47_O_11_); peak 142 as norsolorinic acid (C_20_H_17_O_7_); peak 148 as heptahydroxydioxohexacosanoic acid (C_26_H_47_O_11_); peak 151 as dihydroxytetracosahexaenoic acid (C_24_H_35_O_4_); and peak 152 as hydroxyoctadecadienoic acid (C_18_H_31_O_3_). In the same manner, peak 161 and 162 were attributed to pentadecatetraenoic acid and 9-hydroxyoctadecatrienoic acid, respectively. Finally, peak 171 was identified as trihydroxyheptacosa pentaenoic acid; peak 173 as hydroxytrioxodocosanoic acid; and peak 183 as dihydroxyicosahexaenoic acid (C_20_H_27_O_4_).

In this study, we worked on 11 species of the genus *Sticta* from Colombia and Chile. It should be noted that the species were collected in different ecosystems and environmental conditions in South America. The analyses includes 189 compounds, 41 of which had not yet been identified, of which the most representative are gluconic acid (1), citric acid (5), 2-Ethylisophthalic acid (13), orsellinic acid (34), lecanoric acid (89), stictic acid (105), parietin (106), gyrophoric acid (123) and usnic acid (184) (Figure 3). It should be noted that none of the identified and unidentified compounds are present simultaneously in all 11 species. 

#### 2.1.6. Distance and Phylogenetic Analysis

We found that 69/189 (37%) chemical characters and 7/16 (44%) morphological characters were parsimony informative. Optimally retained trees had a minimum parsimony score of 256 (chemical compounds) and 87 (morphology). The chemical compounds tree showed higher consistency and retention indexes and lower homoplasy (CI = 0.966, RI = 0.786) than the morphology tree (CI = 0.738., RI = 0.531).

In general, maximum-parsimony strict consensus trees from morphological and chemical characters were not consistent with each other and did not recover the same relationship between taxa (Figure 4), whereas the morphological tree recovers most of the evolutionary relationships (positions in phylogeny) documented in the published molecular phylogeny of Colombian *Sticta* [50] and the chemical-compound tree mirrors the geographic clusters of collected samples exactly.

In the last two decades, the study of secondary metabolites in lichens has represented an input for the determination of specimens in different complex groups, through their intervention in taxonomic keys. These compounds are mostly aromatic derivatives such as depsides, depsidones, dibenzofurans, dibenzoquinones and usnic acid among others that derive from the biochemical pathways generated by malonic, mevalonic and shikimic acids [51]. The morphological data used in the present analysis of phylogenetic relationships within species of *Sticta* demonstrate the relevance of morphological traits in lichen taxonomy. Nevertheless, the chemical characters offer the possibility of an alternative comparison, independent from the morphology-based classification system [52]. In this study, the analysis of chemical traits recovered more geographic than ancestor-descendant relationships among taxa. These results enrich the discussion of the role of the local environment on lichen adaptation through the actions of natural selection on biochemical pathways.

In other studies, groupings based on chemical compounds such as the case of *S.* cf *ocaniensis*, *S.* cf. *pseudolobaria*, and *S. canariensis*, were consistent with the known molecular phylogeny of *Sticta*, as was the case with *S*. *pulmonarioides* and *S.* cf. *weigelia* [53]. For the genus *Cetrelia*, an assessment of the composition of secondary metabolites allowed for a confirmation of the presence of species only reported in America in Europe, as in the case of *C. chicitae*. Metabolite composition has also facilitated the confirmation of new species for the genus by chemical fingerprinting, which contrasts the phenotypic plasticity of some morphological characters used for identification [54]. In the genus *Psoroma*, an analysis of the distribution of secondary metabolites has validated the presence of chemical markers unique to the group, and the presumed description of subgenera by taxa heterogeneity, causing spatial segregation [55].

On the other hand, in the genus *Cladonia*, there is evidence of the use of chemotaxonomic methods to determine and differentiate phylogenetically related species (*C. arbuscula*, *C. borealis*, *C. chlorophaea*, *C. coccifera*, *C. coniocraea*, *C. cornuta*, *C. fimbriata*, *C. mitis*, *C. monomorpha*, *C. pyxidate*, *C. rangiferina*, *C. stellaris*, and *C. stygia*), which contain chemical markers exclusive to the group [56]. In this way, the morphological–anatomical data are complemented, and discriminatory characters are provided to distinguish the species. In the genus *Blastenia*, reduced chemotypes are also reported in some lineages with particular genetic characteristics and distribution [57]. 

Currently, the process of chemotaxonomic discrimination analysis in lichen groups requires reinforcement with complementary techniques, such as the use of pigments derived from anthraquinone-type compounds in specimens of the family Teloschistaceae (*Pyrenodesmia* sensu lato) [58] and optical-sensor profiles for metabolic profiling in species of the genera *Cladonia*, *Stereocaulon*, *Lichina*, *Collema* and *Peltigera* [59]. In addition, advances in analytical chemistry and mass spectrometry have allowed for a greater specificity in the elaboration of bioactive compounds profiles, which, together with modern DNA-sequencing techniques and the extension of morphological descriptions as a “polyphasic approach”, provide objectivity in the delimitation of lichen species [60,61]. However, due to the wide variation in lichen chemotypes, the use of new compounds such as fatty acids is proposed in chemotaxonomy and phylogeny analyses, and in building an understanding of molecular-complex communication and compound biosynthetic pathways [62].

## 3. Materials and Methods 

### 3.1. Chemicals 

Ultrapure water was obtained from a water purification system brand Millipore (Milli-Q Merck Millipore, Santiago, Chile). Analytical reagents were all purchased from Sigma Aldrich Co. (Santiago, Chile). Ethanol, Methanol, formic acid, acetone, and acetonitrile were of chromatographic grade for HPLC analysis. Analytical lichen standards (purity: 98% by HPLC) were purchased from Sigma-Aldrich Chemical Company (Santiago, Chile).

### 3.2. Lichen Material

The lichen specimens *S. pseudosylvatica* Moncada & Suárez (35 g) and *S. luteocyphellata* Moncada & Lücking (28 g) were collected by Olimpo García Beltrán and Alfredo Torres Benítez in 2018 in Villahermosa, in the department of Tolima-Colombia, at the farm La Estrella (5°02′48.63″ N–75°07′37.98″ W). The species *S.* cf. *andina* Moncada & Lücking (31 g), *S.* cf. *hypoglabra* Moncada & Lücking (42 g), *S. cordillerana* Gyeln (37 g), *S.* cf. *gyalocarpa* (Nyl.) (29 g), *S. leucoblepharis* (Nyl.) Tuck. & Mont (34 g). y *S. parahumboldtii* Moncada & Lücking (40 g) were collected by Alfredo Torres Benítez and Emmanuel Campos in 2018 in the Semillas de Agua Civil Society Nature Reserve in the Anaime páramo (4°15′18.09″ N–73°33′23.27″ W) and the species *S. impressula* (Nyl.) Zahlbr (29 g) and *S. ocaniensis* (33 g) Moncada & Simijaca were collected by Alfredo Torres Benítez and María Rivera Montalvo in 2017 in the “Combéima river basin”, Ibagué-Tolima, Colombia (4°36′02.35″ N–75°19′50.45″ W). All voucher specimens were deposited in the herbarium of Universidad Distrital Francisco José de Caldas (Colombia) and Prof. Alejandra Suárez Corredor confirmed their identity.

### 3.3. Preparation of the Sample for Analyses

Fresh samples were weighed and frozen for two days at −80 °C. Then, the samples were taken to a freeze–evaporation system (Model 7670541 FreeZone 2.5 Liter Labconco Freeze Dry Systems) and all the water contained in the original product was removed by freeze–evaporation cycles. A total of 3 g of each dried lichen was macerated with methanol (3 times, 30 mL each time, 3 days/extraction). The solutions were concentrated to obtain 11 mg of extract from *S. pseudosylvatica*; 9 mg *S. luteocyphellata*; 14 mg *S.* cf. *andina*; 12 mg *S.* cf. *hypoglabra*; 10 mg *S. cordillerana*; 9 mg *S.* cf. *gyalocarpa*; 13 mg *S. leucoblepharis*; 13 mg *S. parahumboldtii*; 8 mg *S. impressula* and 9 mg *S. ocaniensis*, respectively. Then, the lichen extracts were processed individually for HPLC-MS analyses (redissolved in methanol at a concentration of 1 mg/mL for the analyses).

### 3.4. Instrument

A Thermo Scientific Ultimate 3000 UHPLC with a PDA (photodiode array detector) detector controlled by Chromeleon 7.2 Software (Thermo Fisher Scientific, Waltham, MA, USA) in conjunction with a Thermo high resolution Q-Exactive focus mass spectrometer (Thermo, Bremen, Germany) were used for analysis. The chromatographic system was coupled to the MS using a type II heated electrospray ionization source. Nitrogen obtained (purity >99.999%) from a nitrogen generator (Genius NM32LA, Peak Scientific, Billerica, MA, USA) was employed as both the collision and damping gas. Mass calibration for Orbitrap was performed once a day, in both negative and positive modes, to ensure working mass 5 ppm of accuracy. Sodium dodecyl sulfate, caffeine, N-butylamine, buspirone hydrochloride, and taurocholic acid sodium salt (Sigma Aldrich, Saint Louis, MO, USA) plus Ultramark 1621 (Alpha Aezar, Stevensville, MI, USA), a phosphazine fluorinated solution, was the standard mixture used to calibrate the mass spectrometer. These compounds were dissolved in a mixture of acetic acid, acetonitrile, water, and methanol (Merck, Darmstadt, Germany) and were infused using a Chemyx Fusion 100 syringe pump, XCalibur 2.3 software and Trace Finder 3.2 (Thermo Fisher Scientific, San José, CA, USA), which were used for control and data processing. Q Exactive 2.0 SP 2 from Thermo Fisher Scientific was used to control the mass spectrometer. The lichens extracts were individually redissolved in methanol (at a concentration of 1 mg/mL), each solution was filtered (PTFE filter, Merck) and then 10 microliters were injected in the UHPLC instrument for UHPLC-MS analysis. XCalibur 2.3 software (Thermo Fisher Scientific, Bremen, Germany) and Trace Finder 3.2 (Thermo Fisher Scientific, San José, CA, USA) were used for UHPLC control and data processing, respectively. Q Exactive 2.0 SP 2 from Thermo Fisher Scientific was used to control the mass spectrometer.

### 3.5. LC Parameters

Liquid chromatography on a UHPLC C-18 column (Acclaim, 150 mm × 4.6 mm ID, 2.5 μm, Thermo Fisher Scientific, Bremen, Germany) was performed as reported previously. The mobile phases were 1% formic aqueous solution, (A) methanol 1% formic acid (B) and acetonitrile 1% formic acid (C). The gradient program time were as follows: 0.00 min, 18 B, 75 C; 5.00 min, 18 B, 75 C; 15.00 min, 40 B, 60 C; 20.00 min. B, 100 C; and 12 min for column equilibration at starting conditions.

### 3.6. MS Parameters

The HESI parameters were as follows: sheath gas-flow rate of 75 units; aux. gas unit flow rate of 20; capillary temperature of 400 °C; aux gas heater temperature of 500 °C; spray voltage of 2500 V (for ESI−); and S lens RF level of 30. Full scan data in both the positive and negative modes were acquired at a resolving power of 70,000 FWHM (full width half maximum) at *m/z* 200. For the compounds of interest, a scan range of *m*/*z* 100–1000 was chosen; the automatic gain control (AGC) was set at 3 × 10^6^ and the injection time set to 200 ms. Scan-rate was set at 2 scans s^−1^. External calibration was performed using a calibration solution in the positive and negative modes. For confirmation purposes, a targeted MS/MS analysis was performed using the mass inclusion list, with a 30 s time window, with the Orbitrap spectrometer operating both in the positive and negative mode at 17,500 FWHM (*m/z* 200). The AGC target was set to 2 × 10^5^, with the max. injection time of 20 ms. The precursor ions were filtered by the quadrupole, which operates at an isolation window of *m/z* 2. The fore vacuum, high vacuum and ultrahigh vacuum were maintained at approximately 2 mbar, from 10^5^ and below 10^10^ mbar, respectively. Collision energy (HCD cell) was operated at 30 kv. Detection was based on calculated exact mass and on retention time of target compounds, as shown in Table 1. The mass tolerance window was set to 5 ppm for the two modes for most compounds.

### 3.7. Similarity and Phylogenetic Analyses

We carried out a phylogenetic study to analyze if the chemical compounds found in *Sticta* specimens recovered a phylogenetic signal consistent with the current taxonomic relationships in the genera. First, we built two character-state matrices. One matrix included the 189 chemical compounds reported in this paper, and the other encompassed 16 morphological traits (Appendix A). Then, the compounds were coded as binary characters (presence/absence), whereas morphological traits were coded as multistate characters. All character states receive the same weight and were set as unordered. Next, exploratory Neighbor-Joining distance trees were built. After that, we used a maximum parsimony phylogenetic approach to perform an exhaustive search of optimal trees. A maximum of 100 trees were retained after evaluating ca. 34 million trees per matrix. Trees were unrooted given the absence of descriptions of chemical compounds and morphological traits for potential outgroups. Then, the consistency and retention indexes were calculated. Finally, we obtained the strict consensus of optimal trees. All the procedures were performed in PAUP 4a168 for mac.

## 4. Conclusions

Eleven lichens of the *Sticta* genera from two different country zones were phytochemically investigated. More scientific data on chemistry is presented for these interesting lichens that can significantly increase the knowledge and potential for sustainable applications and industrial interest. This valuable natural-product biomass has potential applications in food, medicine, biotechnology, pharmaceuticals, and cosmetics, with many possible applications from food-conserving agents to anticancer biomaterials. The morphological data used in the present analysis of phylogenetic relationships within genus of *Sticta* demonstrate the relevance of morphological traits in lichen taxonomy. Nevertheless, chemical characters for chemotaxonomic studies offer the possibility of an alternative comparison, independent from the morphology-based classification system. In this study, chemical traits’ analysis recovered more geographic than ancestor-descendant relationships among taxa, and these results enriched the discussion of the role of the local environment on lichen adaptation through natural selection acting on biochemical pathways.

## Figures and Tables

**Figure 1 metabolites-12-00156-f001:**
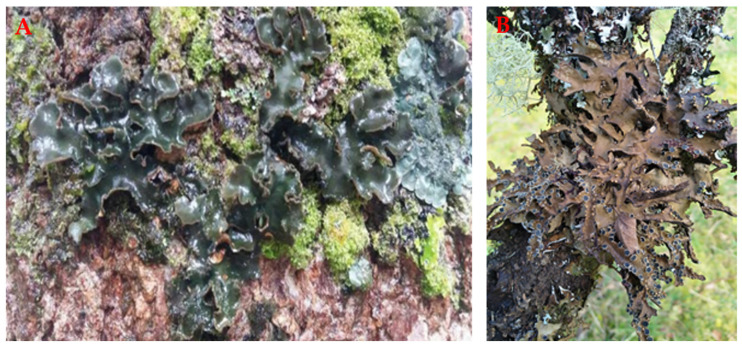
(**A**) *S. pseudosylvatica* (Colombia); (**B**) *S. lineariloba* (Chile).

**Figure 2 metabolites-12-00156-f002:**
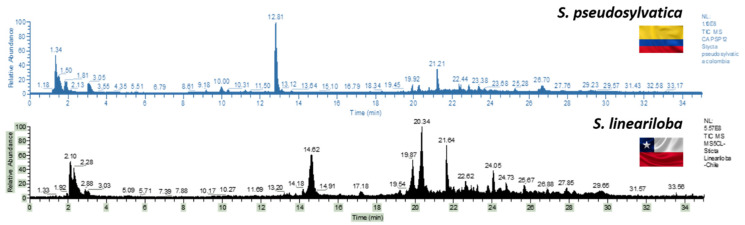
Chromatogram of the species *S. pseudosylvatica* and *S. lineariloba*.

**Figure 3 metabolites-12-00156-f003:**
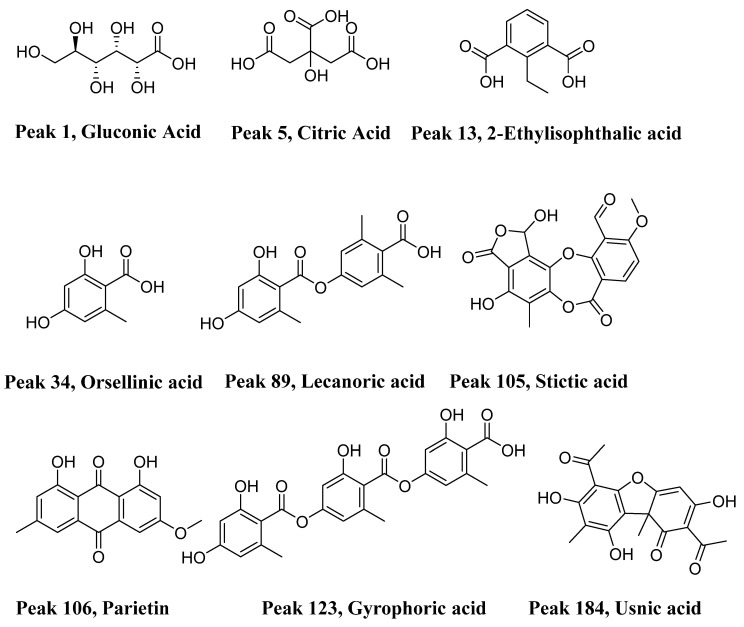
Structures of some representative compounds found in *Sticta* lichens.

**Figure 4 metabolites-12-00156-f004:**
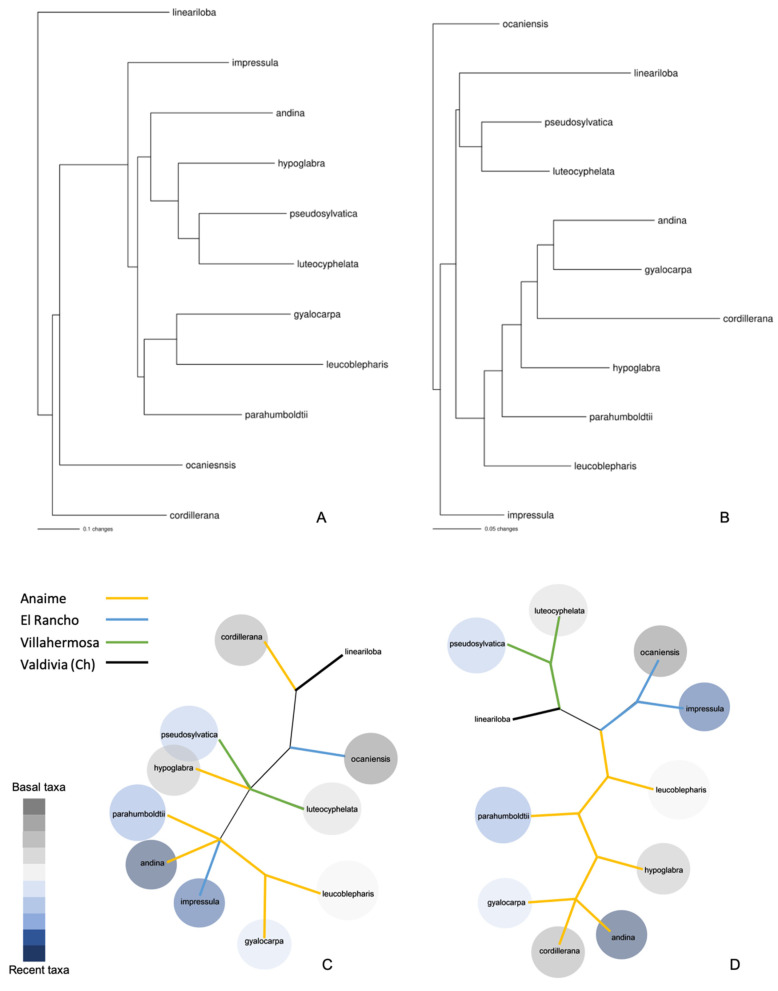
Neighbor joining trees (**A**,**B**) and unrooted strict consensus trees from maximum parsimony analysis (**C**,**D**) with morphological traits (**A**,**C**) and chemical compounds (**B**,**D**) in some Colombian species of *Sticta*. Colored lines represent localities from collected specimens. Colored circles represent their position in the molecular phylogeny of Moncada et al. (2014); grayish colors in the color scale refer to more basal taxa, whereas bluish colors are more recent taxa. Although *S. lineariloba* occurs in Colombia, the specimen in this study was collected in Valdivia (Chile). Independent of the grouping method, trees built based on chemical compounds recovered the geographic relationships, while trees built with morphological traits resemble the molecular phylogeny of Colombian *Sticta*.

**Table 1 metabolites-12-00156-t001:** Identification of metabolites in 11 species of the genus *Sticta* from Colombia and Chile.

Peak	Tentative Identification	[M-H]-	Retention Time (min)	Theoretical Mass (*m/z*)	Measured Mass (*m/z*)	Accuracy (ppm)	Metabolite Type **	MS Ions (*m*/*z*)	Lichen Species *
1	Gluconic Acid	C_6_H_11_O_7_	1.28	195.0509	195.0505	2.0	Acid	165.0401;	2; 3; 7; 9; 10; 11
2	Unknown	C_11_H_5_O_5_N	1.28	231.0184	231.0195	−4.8	-	---	6
3	Manitol	C_6_H_13_O_6_	1.31	181.0712	181.0714	−1.1	Carbohidrate	181.0717; 163.0606	7; 9; 10; 11
4	Arabic acid	C_5_H_9_O_6_	1.29	165.0399	165.0401	−1.2	Acid	147.0293; 113.0237; 129.0196	6
5	Citric Acid	C_6_H_7_O_7_	1.38	191.0196	191.0192	2.0	Acid	111.0080	1; 2; 3; 4; 5; 6; 7; 8; 9; 11
6	Unknown	C_15_H_5_O_3_N_2_	1.38	261.0289	261.0300	−4.2	--	---	1; 7; 8; 10
7	4-ethyl-2-Ethylisophthalic acid	C_10_H_9_O_4_	1.44	193.0504	193.0501	1.6	A	133.0288	1; 2; 3; 4; 6
8	Unknown	C_8_H_10_O_5_N	1.56	200.0563	200.0559	2.0	-	--	1; 3; 4; 6
9	Unknown	C_15_H_5_O_3_N_2_	1.64	261.0247	261.0278	−11.8	-	---	7; 8; 10
10	Isocitric Acid	C_6_H_7_O_7_	1.77	191.0195	191.0192	1.6	Acid	111.0079;	1; 2; 3; 4; 6; 7
11	Unknown	C_7_H_9_O_7_	1.91	205.0352	205.0348	2.0	-	187.0245; 173.0087; 121.1131	2
12	Unknown	C_7_H_7_O_6_	2.24	187.0246	187.0243	1.6	-	125.0237; 137.2503;	2
13	2-Ethylisophthalic acid	C_10_H_9_O_4_	2.85	193.0505	193.0501	2.0	A	161.0240; 133.0290	1; 2; 3; 4; 6; 7; 8; 9; 10
14	Trihydroxy benzaledehyde	C_7_H_5_O_4_	4.81	153.0188	153.0189	−0.7	A	137.0238	1; 2; 3; 4
15	2-Hydroxyisophthalic acid	C_8_H_5_O_5_	7.63	181.0137	181.0141	−2.2	A	137.0238	4
16	2,4-dihydroxy benzaldehyde	C_7_H_5_O_3_	8.02	137.0238	137.0239	−0.7	A	121.0289	1; 2; 3; 4
17	4-O-Demethylglomellic acid	C_24_H_25_O_9_	8.65	457.1476	457.1499	−5.0	d	---	5; 8
18	Unknow	C_22_H_23_O_7_	8.77	399.1444	399.1413	7.8	-	---	8
19	Unknow	C_7_H_11_O_5_	8.86	175.0606	175.0611	−2.9	-	---	5
20	Grayanic acid	C_23_H_25_O_7_	8.91	413.1600	413.1569	7.5	d	181.06503	8
21	Unknown	C_13_H_16_O_7_N	8.98	298.0940	298.0927	4.4	-	175.0609; 283.0210; 202.0696;	5
22	Unknown	C_11_H_9_O_7_	9.33	253.0361	253.0348	5.1	-	137.0603; 181,0505; 249.3808; 173.4203;	5
23	Atranol	C_8_H_7_O_3_	9.44	151.0395	151.0396	−0.7	A	123.0445; 135.0445	3
24	Unknown	C_24_H_23_ O_8_	9.48	439.1371	439.1393	−5.0	-	151.0397; 105.0948; 121.4871;	5
25	5,7-Dihydroxy-6-methylphthalide	C_9_H_7_O_4_	9.64	179.0344	179.0347	−1.6	A	135.0444; 107.0494	4; 5; 7; 8; 10
26	Unknown	C_16_H_15_O_10_	9.73	367.0665	367.0639	7.1	-	---	1; 4
27	Unknown	C_30_H_27_O_13_N	10.03	609.1475	609.1482	−1.1	-	---	1; 4; 6
28	Unknow	C18H15O4	10.31	295.0970	295.0935	11.8	-	---	9
29	Unknow	C_17_H_13_O_6_	11.31	313.0712	313.0724	−3.8	--	---	1; 2; 3; 4; 5
30	1,5-Pentanedicarboxylic acid	C_7_H_11_O_4_	10.56	159.0657	159.0660	−1.9	L	115.0758	5
31	Unknown	C_29_H_25_O_13_N	10.54	595.1316	595.1326	−1.7	-	---	3
32	Didechlorolecideoidin	C_17_ H_13_O_7_	10.64	329.0661	329.0676	−4.6	D	209.0456; 285.0776; 151.0396; 179.0347; 123.0443	3
33	Decahydroxyoxopentacosanoic acid	C_25_H_47_O_13_	10.71	555.3017	555.3047	−5.4	L	---	8
34	Orsellinic acid	C_8_H_7_O_4_	11.04	167.0347	167.0344	1.8	A	123.0442	2; 3; 4; 5; 6
35	Unknow	C_21_H_19_O_12_	11.04	463.0877	463.0893	−3.4	-	---	3
36	Unknow	C_10_H_9_O_5_	11.07	209.0450	209.0458	−3.8	-	---	5; 6
37	Nor 8′-methylconstictic acid	C_21_H_19_O_11_	11.11	447.0927	447.0942	−3.4	d	209.0455	2
38	Unknow	C_17_H_13_O_6_	11.20	313.0712	313.0720	−2.5	-	---	7
39	Metil-2,6-dihidroxibenzoate	C_8_H_7_O_4_	11.21	167.0344	167.0346	−1.1	A	109.0287; 137.0238	9
40	Hypostictic acid isomer	C_19_H_15_O_8_	11.36	371.0778	371.0782	−1.1	D	195.0665; 327.0885; 341.0679; 179.0347	1; 3; 4; 7
41	Unknow	C_19_H_16_O_9_ N	11.69	402.0825	402.0841	−3.98	-	---	11
42	Fumarprotocetraric acid derivative	C_17_H_11_O_6_	11,70	311.0556	311.0564	−2.5	d	---	7; 10
43	4,5-Dihydroxy-2-nonenoic acid	C_9_H_15_O_4_	12.11	187.0974	187.0977	−1.6	L	171.1025; 143.1072	1; 2; 3; 4; 9; 10
44	2,4-Dicarboxy-3-hydroxy-5-methoxytoluene	C_10_H_9_O_6_	12.14	225.0407	225.0399	3.5	A	181.0503; 167.0346; 149.0240	4
45	Unknown	C_17_H_13_O_6_	12.16	313.0724	313.0727	−1.0	-	---	4
46	Unknown	C_18_H_15_O_7_	12.37	343.0818	343.0826	−2.3	-	---	7; 9
47	Unknown	C_21_H_17_O_12_	12.40	461.0737	461.0720	3.7	-	---	6
48	Unknown	C_20_H_17_O_8_	12.52	385.0939	385.0923	4.1	-	---	1; 2; 3
49	Unknow	C_10_H_9_O_4_	12.81	193.0501	193.0502	−0.5	-	---	1; 2; 3; 4; 5; 6; 7; 8; 9; 10
50	Unknow	C_27_H_45_O_6_	13.04	465.3216	465.3231	−3.2	-	---	3
51	2,4-dihydroxy benzaldehyde	C_7_H_5_O_3_	13.07	137.0237	137.0239	−1.5	A	121.0288	5
52	Consalizinic acid derivative I	C_19_H_13_O_11_	13.36	417.0458	417.0474	−3.8	D	373.0573; 387.0373; 225.0406, 177.0193	11
53	4-Ethoxy-3-formyl-2-hydroxy-6-methylbenzoic acid	C_11_H_11_O_5_	13.41	223.0614	223.0606	3.6	A	177.0190; 133.0296;	1; 2; 3; 4; 7; 9; 10
54	Unknow	C_20_H_15_O_8_	13.47	383.0767	383.0781	−3.6	-	---	1
55	Consalizinic acid derivative II	C_20_H_17_O_11_	13,47	433.0771	433.0787	−3.7	D	401.0524; 417.0474; 373.0574	11
56	Cynodontin or Citreorosein isomer,	C_15_H_9_O_6_	13.71	285.0399	285.0410	−3.9	Anthraquinone	151.0396; 137.0237	2
57	Consalizinic acid derivative I isomer	C_19_H_13_O_11_	13.78	417.0458	417.0474	−3.84	D	373.0573; 343.0467, 77.0190; 401.0523	10
58	Unknow	C_30_H_47_O_7_	13.79	519.3322	519.3337	−2.9	-	---	3
59	1,4,5,6,8-Pentahydroxy-3-ethylanthraquinone	C_15_H_9_O_7_	13.87	301.0348	301.0361	−4.3	Anthraquinone	---	4
60	Unknow	C_19_H_15_O_4_	13.87	307.0970	307.0939	10.0	-	---	8
61	Unknow	C_14_H_13_O_7_	13.99	293.0661	293.0674	−4.4	-	---	3
62	Haemathamnolic acid isomer	C_19_H_15_O_10_	14.18	403.0665	403.0681	−3.97	D	359.0781; 371.0414; 209.0455	11
63	Fumarprotocetraric acid derivative	C_17_H_11_O_6_	14.78	311.0567	311.0569	−0.6	d	---	1; 2; 3; 7; 10
64	Constictic acid	C_19_H_13_O_10_	14.61	401.0509	401.0528	−4.74	D	357.0625; 313.0726; 283.0619; 255.0670; 121.0289	11
65	Hypostictic acid isomer	C_19_H_15_O_8_	14.98	371.0781	371.0767	3.8	D	327.0883; 195.0664; 179.0347	5
66	Thelephoric acid	C_18_H_7_O_8_	14.81	351.0141	351.0154	−3.7	Terphenylquinones	---	3
67	Methylstictic acid	C_20_H_15_O_9_	15.26	399.0716	399.0728	−3.0	D	371.0779; 193.0504	2; 7
68	Nor 8′-metilconstictic acid	C_21_H_19_O_11_	15.28	447.0927	447.0944	−3.80	D	401.0524; 209.0455	11
69	Protocetraric acid	C_18_H_13_O_9_	17.17	373.0560	373.0574	−3.75	D	355.0468; 329.0674; 311.0568; 227.0352; 267.0669; 285.0777	11
70	Hypoconstictic acid	C_19_H_15_O_9_	17.27	387.0716	387.0729	−3.4	D	267.0673; 311.0552; 149.0238; 343.0827;167.0345	3
71	Unknow	C_14_H_13_O_6_	17.84	277.0712	277.0724	−4.3	-	--	3; 6
72	12,13,15-Trihydroxy-9-octadecenoic acid	C_18_H_15_O_5_	18.11	329.2328	329.2340	−3.6	L	285.1716;	2; 6
73	Unknow	C_20_H_13_O_8_	18.19	381.0610	381.0626	−4.2	-	---	3; 4
74	Salazinic acid	C_18_H_11_O_10_	18.21	387.0352	387.0368	−4.13	D	343.0468; 269.0458; 241.0507; 325.0365; 299.0569	11
75	Unknown	C_10_H_9_O_4_	18.45	193.0505	193.0501	2.1	-	---	3; 4
76	Unknown	C_30_H_47_O_7_	18.61	519.3322	519.3319	0.6	-	--	3
77	Menegazziaic acid	C_18_H_13_O_9_	18.79	373.0560	373.0575	−4.0	D	311,0570; 255.0666; 329.0679	3
78	Norstictic acid	C_18_H_11_O_9_	18.86	371.0403	371.0417	−3.8	D	327.0526; 151.0396; 123.0444;	3
79	Unknow	C_22_H_19_O_10_	18.88	443.0978	443.0996	−4.1	-	---	3
80	Physodalic acid	C_20_H_15_O_10_	18.99	415.0665	415.0681	−3.85	D	359.0417; 315.0520; 343.0832; 387,0367; 373.0573; 401.0525	11
81	Unknow	C_26_H_19_O_10_	19.00	491.0978	491.0997	−3.9	-	---	3
82	Derivative methyl 8-hydroxy-4-0-demethylbarbatate	C_19_H_19_O_9_	19.05	391.1045	391.1029	4.1	d	359.0788	3
83	12,13,15-Trihydroxy-9-octadecenoic acid	C_18_H_33_O_5_	19.05	329.2328	329.2336	−2.4299	L	---	7
84	Haemoventosin	C_15_H_11_O_7_	19.16	303,0519	303.0505	4.7	Naphthaquinone	259.0619; 231.0667; 189.0560;	3
85	α-acetilconstictic acid derivative I	C_21_H_17_O_11_	19.22	445.0771	445.0786	−3.3	D	415.0680; 371.0780; 427.0676; 343.0830;193.0504, 401.0522	11
86	Conhypoprotocetraric acid or Convirensic acid	C_18_H_15_O_8_	19.25	359.0781	359.0767	3.9	D	344.0545; 302.0442	3
87	4-O-dimethylbaemycesic acid	C_18_H_15_O_8_	19.27	359.0781	359.0767	3.9	d	181.0714; 163.0397; 137.0236	1; 2; 3;5; 6
88	Orsellinic acid Isomer	C_8_H_7_O_4_	19.45	167.0344	167.0346	−1.1974	A	123.0440; 149.0235	9; 10
89	Lecanoric acid	C_16_H_13_O_7_	19.51	317.0661	317.0671	0.6	d	167.0345; 123.0443; 149.0238;	1; 2; 3; 4; 5; 6
90	Constictic acid isomer	C_19_H_13_O_10_	19.56	401.0509	401.0524	−3.74	D	357,0626; 313.0726; 343.0831; 255.0622	11
91	Pentahydroxytetracosanoic acid	C_24_H_47_O_7_	19.67	447.3322	447.3336	−3.1	L	---	1; 3; 7; 9; 10
92	2-Methyl-5-hydroxy-6-hydroxymethyl-7-Methoxychromone	C_12_H_11_O_5_	19.73	235.0606	235.0615	−3.8	C	181.0504	3
93	Unknown	C_20_H_17_O_8_	19.79	385.0939	385.0923	4.1	-	---	1; 4; 5; 10
94	Heptahydroxytrioxooctadecanoic acid	C_18_H_29_O_12_	19.86	437.1664	437.1645	4.3	L	---	1; 4; 5; 6; 7; 9; 10
95	5,7-Dihydroxy-6-methylphthalide derivative	C_9_H_7_O_3_	19.88	163.0395	163.0392	1.8	A	119.0492	10
96	Criptostictic acid derivative	C_18_H_11_O_8_	20.04	355.0454	355.0462	−2.2	D	133.0288; 239.0715; 311.0572; 179.0345;	7
97	Unknow	C_18_H_17_O_6_	20.08	329.1025	329.1032	−2.1	-	---	9
98	Unknow	C_20_H_15_O_8_	20.16	383.0767	383.0775	−2.0	-	---	9; 10
99	Unknow	C_21_H_19_O_9_	20.18	415.1045	415.1029	3.9	-	---	3; 5; 9; 10
100	Unknow	C_28_H_23_O_11_	20.12	535.1240	535.1257	−3.1	-	---	3
101	Unknow	C_15_H_13_O_3_	20.13	241.0872	241.0874	−0.8	-	----	1; 2
102	Heptahydroxytetraoxoicosanoic acid	C_20_H_31_O_13_	21.19	479.1765	479.1746	3.9	L	---	7
103	Tetrahydroxytricosanoic acid	C_23_H_45_O_6_	20.26	417.3232	417.3216	3.9	L	403.3073	1; 3; 4; 7
104	Tetrahydroxytrioxoundecanoic acid	C_11_H_15_O_9_	20.30	291.0716	291.0699	5.8	L	---	8
105	Stictic acid	C_19_H_13_O_9_	20.34	385.0560	385.0576	−4.1	D	341.0674; 357.0622; 297.0774; 313.0721; 193.0504; 269.0826	11
106	Parietin	C_16_H_11_O_5_	20.39	283.0606	283.0617	−3.9	Antraquinone	179.0345	1; 2; 6
107	Unknow	C_24_H_47_O_11_N_2_	20.39	539.3157	539.3180	−4.3	-	---	3
108	Evernic acid isomer	C_17_H_15_O_7_	20.46	331.0818	331.0830	−3.6	d	167.0347; 123.0447; 149.0240	1
109	Hypoconstictic acid	C_19_H_15_O_9_	20.50	387.0716	387.0732	−4.1	D	149.0238; 343.0836; 167.0345	4
110	Cryptostictic acid	C_19_H_15_O_9_	20.50	387.0716	387.0725	−2.3	D	267,0661; 343,0825; 311.05067; 239.0710	7; 8
111	Retigeric acid derivative	C_30_H_43_O_7_	20.51	515.3009	515.3025	−3.1	Triterpene	---	3
112	Retigeric acid B	C_30_H_45_O_6_	20.56	501.3216	501.3236	−4.0	Triterpene	---	3
113	Salazinic acid isomer	C_18_H_11_O_10_	20.58	387.0352	387.0368	−4.13	D	343.0468; 299.0565	11
114	Unknown	C_23_H_22_O_10_N	20.68	472.1244	472.1259	−3.2	-	-----	6
115	9,10-dihydroxyoctadecatrienoic acid	C_18_H_29_O_4_	20.69	309.2081	309.2066	4.9	L	291.1975	1; 2; 4
116	Unknow	C_17_H_13_O_6_	20.84	313.0712	313.0727	−4.8	-	----	1; 2; 6
117	Pulvinic acid derivative I	C_18_H_11_O_6_	20.97	323.0556	323.0556	0.0	Pulvinic acid y derivates	133.0286; 117.0335	10
118	9,10,12 Trihydroxytriacontaheptaenoic acid	C_30_H_45_O_5_	20.99	485.3284	485.3267	3.5	L	---	3
119	4-0-Demethylbarbatic acid	C_18_H_17_O_7_	20.99	345.0974	345.0989	−4.3	d	181.0505; 163.0396; 137.0603	4
120	Unknow	C_24_H_23_O_10_N	21.02	485.1322	485.1319	0.6	-	---	1; 5
121	Methyl orsellinate	C_9_H_9_O_4_	21.05	181.0502	181.0501	0.5	A	163.0389	1
122	Heptahydroxyetraoxoicosanoic acid	C_20_H_31_O_13_	21.17	479.1752	479.1765	−2.7	L	---	1; 2; 3; 4; 5; 6; 9
123	Gyrophoric Acid	C_24_H_19_O_10_	21.25	467.0991	467.0978	2.8	d	167.0346; 317.0673; 123.0445; 149.0238;	1; 3; 4; 6; 11
124	Galbinic acid	C_20_H_13_O_11_	21.27	429.0458	429.0474	−3.73	D	403.0681; 371.0417; 401.0524; 327.0518; 149.0239	11
125	Hyposalazinic acid	C_18_H_13_O_8_	21.28	357.0610	357.0623	−3.6	D	313.0723; 135.0444; 179.0348	1
126	Hydroxytetracosapentaenoic acid	C_24_H_37_O_3_	21.42	373.2743	373.2743	0.0	L	---	10
127	Orsellinic acid isomer	C_8_H_7_O_4_	21.47	167.0347	167.0344	1.8	A	149.0239; 123.0443	1; 4; 6
128	Dihydroxyoctadecenoic acid	C_18_H_33_O_4_	21.48	313.2390	313.2395	−1.6	L	----	2; 5; 6
129	Norstictic acid	C_18_H_11_O_9_	21.64	371.0403	371.0417	−3.77	D	27.0517; 227.0716; 151.0390; 243.0297	11
130	Dihydroxyoctadec-6-enoic acid	C_18_H_33_O_4_	21.58	313.2379	313.2379	0.0	L	---	10
131	Loxodinol isomer	C_25_H_29_O_9_	21.64	473.1812	473.1818	−1.2	DE	429.1919	9
132	EthyI 2,4-dihydroxy-6-n-nonylbenzoate	C_18_H_27_O_4_	21.65	307.1909	307.1922	−4.2	A	263.1659	1; 2; 3; 4
133	Evernic Acid	C_17_H_15_O_7_	21.81	331.0828	331.0818	3.0	d	167.0345; 123.0444; 149,0238;	1; 2; 3; 4; 5
134	Protocetraric acid Isomer	C_18_H_13_O_9_	21.85	373.0560	373.0573	−3.48	D	355.0460; 329.0674; 285.0780, 311.0567; 255.,0672	11
135	Unknown	C_22_H_22_O_8_N	21.83	428.1360	428.1345	3.5	-	----	6
136	Strepsilin	C_15_H_9_O_5_	21.89	269.0450	269.0462	−4.5	DBF	225.0554	2; 5
137	Unknown	C_30_H_29_O_4_	21.97	453.2066	453.2061	1.1	-	---	3
138	Unknown	C_18_H_11_O_6_	22.01	323.0556	323.0570	−4.33	-	---	11
139	Hexahydroxytrioxooctacosatrienoic acid	C_28_H_43_O_11_	22.12	555.2805	555.2841	−6.4832	L	---	9
140	Nonahydroxyoctacosatetraenoic acid	C_28_ H_47_O_11_	22.26	559.3124	559.3132	−1.4	L	---	2
141	Unknow	C_28_H_41_O_9_N_2_	22.26	549.2849	549.2812	6.8	-	---	3
142	Norsolorinic acid	C_20_H_17_O_7_	22.44	369.0974	369.0989	−4.06	-	---	11
143	Unknow	C_25_H_33_O_13_	22.46	541.1921	541.1909	2.2	-	---	1; 2; 3; 6; 10
144	Hydroxytetracosapentaenoic acid derivative	C_24_H_37_O_3_	22.61	373.2743	373.2741	0.5	-	---	10
145	Hydroxytrioxotricosanoic acid	C_23_H_39_O_6_	22.53	411.2747	411.2757	−2.4	L	---	8
146	Squamatic acid	C_19_H_17_O_9_	22.89	389.0873	389.0886	−3.3	d	343.0836; 163.0396; 193.0139; 149.0238; 121.0286	1; 3; 4
147	Picrolichenic acid	C_25_H_29_O_7_	22.72	441.1913	441.1926	−3.0	Depsones	---	1
148	Heptahydroxydioxohexacosanoic acid	C_26_H_47_O_11_	22.76	535.3118	535.3134	−3.0	L	---	6
149	Unknow	C_30_H_45_O_4_	22.83	469.3319	469.3335	−3.4	-	---	3
150	2,2′-Di-O-methylanziaic acid	C_26_H_33_O_7_	22.85	457.2226	457.2244	−3.9	d	413.2345;	4
151	Dihydroxytetracosahexaenoic acid	C_24_H_35_O_4_	22.85	387.2535	387.2552	−4.4	L	---	5
152	Hydroxyoctadecadienoic acid	C_18_H_31_O_3_	22.90	295.2273	295.2273	0.0	L	---	10
153	Orsellinic acid Isomer	C_8_H_7_O_4_	22.92	167.0344	167.0348	−2.3	A	149.0240; 123.0445	11
154	Pulvinic acid	C_18_H_11_O_5_	22.98	307.0606	307.0613	−2.2	Pulvinic acid y derivates	117.0338; 263.0713	9
155	4-0-Demethylbarbatic acid	C_18_H_17_O_7_	23.02	345.0974	345.0986	−3.5	d	123.0443; 137.0237; 181.0502	1
156	Psoromic acid	C_1__8_H_13_O_8_	23.06	357.0610	357.0626	−4.4	D	313.0726; 181.0502; 179.0347; 327.0520; 269.0826; 285.0776	11
157	Methylgyrophoric acid	C_25_H_21_O_10_	23.15	481.1135	481.1147	−2.5	d	149.0238; 123.0442; 167.0346; 317.0671	1; 4
158	Evernic acid isomer	C_17_H_15_O_7_	23.22	331.0818	331.0832	−4.2	d	149.0239; 123.0443; 167.0346; 105.0337	11
159	Skyrin	C_30_H_17_O_10_	23.28	537.0822	537.0840	−3.4	Anthraquinones	----	3
160	Angardianic acid	C_19_H_35_O_4_	23.36	327.2543	327.2547	−1.2	Acids	283.2649; 309.2081	2; 4
161	Pentadecatetraenoic acid	C_15_H_21_O_2_	23.38	233.1542	233.1545	−1.2	L	---	9; 10
162	9-hydroxyoctadecatrienoic acid	C_18_ H_29_O_3_	23.45	293.2117	293.2130	−4.4	L	277.2180	6
163	Unknow	C_18_H_15_O_7_	23.53	343.0818	343.0824	−1.7	-	---	9
164	Pulvinic acid derivative II	C_19_H_13_O_5_	23.68	321.0763	321.0770	−2.1	Pulvinic acid y derivates	117.0337	9; 10; 11
165	Pulvinic acid	C_18_H_11_O_5_	23.77	307.0606	307.0620	−4.5	Pulvinic acid y derivates	263.0720; 117.0339	11
166	Furfuric acid isomer	C_28_H_23_O_12_	23.82	551.1190	551.1197	−1.2	D	371.0784; 193.0504; 179.0347; 207.0297; 193.0504	8
167	Unknow	C_26_H_47_O_5_N_2_	23.82	467.3485	467.3492	−3,9	-	---	3
168	Unknow	C_30_H_27_O_6_	23.89	483.1808	483.1820	−2,5	-	---	1
169	Unknow	C_30_H_25_O_6_	24.01	481.1651	481.1663	−2,5	-	---	1
170	Unknow	C_15_H_13_O_3_	24.02	241.0872	241.0872	0,0	-	---	1
171	Trihydroxyheptacosa pentaenoic acid	C_27_H_43_O_5_	24.05	447.3110	447.3127	−3.8	L	---	8
172	Barbatic Acid	C_19_H_19_O_7_	24.26	359.1141	359.1131	2.8	d	137.0603; 163.0396; 181.0509	1; 4
173	Hydroxytrioxodocosanoic acid	C_22_H_37_O_6_	24.29	397.2590	397.2601	−2.8	L	---	8
174	Thamnolic acid isomer	C_19_H_15_O_11_	24.41	419.0614	419.0630	−3.8	d	375.0730; 167.0344; 209.0455; 181.0503	11
175	Orsenillic acid derivated II	C_8_H_7_O_4_	24.73	167.0344	167.0347	−1.80	-	149.0239; 1230444;	11
176	Unknow	C_26_H_33_O_8_	24.73	473.2190	473.2175	3.2	-	---	2
177	Lobaric acid	C_25_H_27_O_8_	24.81	455.1706	455.1718	−2.6	D	411.1824; 367.1811	1
178	Unknow	C_22_H_27_O_7_	24.97	403.1770	403.1757	3.2	-		2
179	Unknow	C_30_H_41_O_8_	25.27	529.2819	529.2801	3.4	-	---	4
180	Hypothamnolic acid	C_19_H_17_O_10_	25.49	405.0822	405.0832	−2.5	d	209.0456; 181.0499	1
181	Unknow	C_25_H_11_O_7_	25.53	423.0505	423.0497	1.9	-	---	4
182	Pulvinic acid derivative III	C_19_H_13_O_5_	25.67	321.0763	321.0777	−4.3	Pulvinic acid y derivates	117.0338	11
183	Dihydroxyicosahexaenoic acid	C_20_H_27_O_4_	26.02	331.1909	331.1925	−4.8	L	-----	11
184	Usnic acid	C_18_H_15_O_7_	26.05	343.0818	343.0831	−3.8	DBF	231.0658; 328.0585; 259.0604	3; 4; 6; 7; 8
185	Nephromopsic acid orRoccellaric acid	C_19_H_33_O_4_	26.32	325.2392	325.2379	4.0	Acids	281.2494	3; 4
186	Unknow	C_28_H_25_O_5_N	26.87	455.1733	455.1723	2.20	-	---	11
187	Perlatolic acid	C_25_H_31_O_7_	26.98	443.2070	443.2078	−1.8	d	205.0867; 179.1073; 223.0973	7; 8
188	Caperatic acid	C_21_H_37_O_7_	28.14	401.2539	401.2549	−2.4	Acids	255.2327	8
189	Atranorin	C_19_H_17_O_8_	29.64	373.0923	373.0937	−3.75	d	177.0192; 163.0397	9; 11

* Identified by addition experiments with a genuine compound. A = Aromatic compound; L = Lipid; D = depsidone; d = depside; DE = diphenilether; DBF = dibenzofurane. C = Chromone. **** 1** = *S.* cf. *andina**a*; **2** = *S.* cf. *hypoglabra*; **3**
*= S. cordillerana*; **4**
*= S.* cf. *gyalocarpa*; **5*** = S. leucoblepharis*; **6**
*= S. parahumboldtii*; **7** = *S. impressula*; **8** = *S. ocaniesnsis*; **9** = *S. speudosylvatica*; **10** = *S. luteocyphelata*; **11** = *S. lineariloba*.

## Data Availability

Data is contained within the article or Appendix A, but raw Thermo HPLC profiles of the plant or other data can be available on author’s request.

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
