# Peer review of "Phylogenetic Studies and Metabolite Analysis of Sticta Species from Colombia and Chile by Ultra-High Performance Liquid Chromatography-High Resolution-Q-Orbitrap-Mass Spectrometry"

_metabolites, 2022, doi:10.3390/metabo12020156_

Round 1
Reviewer 1 Report
This paper is interesting and important taking into account the possible application of the investigated lichens. The idea of the work is good and it is well done. But I have some remarks which should be clarified before making decision concerning the publication of the paper.
Some acronyms are clarified, e. g. HCD (l. 68) and some are not, e. g. DAD (l. 64). The same type of detector used in this paper is described as PDA without clarification – l. 304. And what for this detector was exactly used in this paper?
The sentence in l. 73 – 78 is too long and therefore is unclearly.
In table 1 instead of theorical should be theoretical.
L 298 – 302. The information is not sufficient. Solution was concentrated and some amounts of substance was obtained. Is it the amount of concentrated solution or rather the amount of dry substance? In such situation it would be not concentration but the evaporation of the solvent to dryness. The information concerning extracts in l. 318 is connected with that.
L 325. What does mean 2.5 m?
L 327 – 329. The description of mobile phase gradient is unintelligible. I do not understand it at all.
L 367. One offer should be deleted.
Author Response
Reviewer 1
Comments and Suggestions for Authors
This paper is interesting and important taking into account the possible application of the investigated lichens. The idea of the work is good and it is well done. But I have some remarks which should be clarified before making decision concerning the publication of the paper.
Some acronyms are clarified, e. g. HCD (l. 68) and some are not, e. g. DAD (l. 64). The same type of detector used in this paper is described as PDA without clarification – l. 304. And what for this detector was exactly used in this paper?
Answer: Dear reviewer, we appreciate the suggestions made for the improvement of the manuscript, the changes have been made in the manuscript.
The sentence in l. 73 – 78 is too long and therefore is unclearly.
Answer: Dear reviewer, we appreciate the suggestions made for the improvement of the manuscript, the changes have been made in the manuscript.
In table 1 instead of theorical should be theoretical.
Answer: Dear reviewer, we appreciate the suggestions made for the improvement of the manuscript, the changes have been made in the manuscript.
L 298 – 302. The information is not sufficient. Solution was concentrated and some amounts of substance was obtained. Is it the amount of concentrated solution or rather the amount of dry substance? In such situation it would be not concentration but the evaporation of the solvent to dryness. The information concerning extracts in l. 318 is connected with that.
Answer: Dear reviewer, we appreciate the suggestions made for the improvement of the manuscript, the changes have been made in the manuscript.
L 325. What does mean 2.5 m?
Answer: Dear reviewer thank you for showing the typo, we refer to u, and the changes have been made in the writing.
L 327 – 329. The description of mobile phase gradient is unintelligible. I do not understand it at all.
Answer: Dear reviewer, we appreciate the suggestions made for the improvement of the manuscript, the changes have been made in the manuscript.
L 367. One offer should be deleted.
Answer: Dear reviewer, we appreciate the suggestions made for the improvement of the manuscript, the changes have been made in the manuscript.

Reviewer 2 Report
Comments for authors
This paper entitled “Phylogenetic studies and Metabolite Analysis of Sticta Species from Colombia and Chile by High Resolution UHPLC-Q-Orbitrap-Mass Spectrometry” by Albornoz et al., reported a work in the metabolomic analysis of 11 species of lichens extracts. The authors claimed that189 tentative analyses or features belong to five groups of compounds based on LC-MS data. In addition, the authors assessed the usefulness of chemical compounds in comparison to traditional morphological traits in the study of ancestor-descendant relationships in the genus. However, several questions should be addressed. The most important is the validation of those metabolites identification from lichens extracts; there is no method validation data (such as repeatability) for the detection of those compounds, so it is hard to evaluate the qualitative results. The authors did not mention how they identify and annotate 189 tentative metabolites. Can they be from a database or published references? Authors should provide evidence to show the credibility and accuracy for the identification of those endogenous metabolites. Some metabolite standards should be available to confirm their identification. The addition of the assessment of the usefulness of chemical compounds in comparison to traditional morphological traits is nice (although the compounds tree could not recover most evolutionary relations, authors claimed it can be used to distinguish the geographic clusters of collected samples.), but the quantitative analysis of those compounds has not been made. The authors carried out a phylogenetic study based on the presence/absence of compounds, however, are there any differences between biologic replicates collected from the same ecosystems and environmental conditions in South America? Further, regarding the LC-MS data analysis, did the authors check and exclude the noise signals from the background? Did the authors check and eliminate the interferences from the in-source fragmentation (e.g., Peak 1 & Peak 4, Peak 23 & Peak 24, and Peak 36 & Peak 37)? Why did the authors only profile metabolites using UHPLC-MS in negative mode?
Lines 3-4, “… High Resolution UHPLC-Q-Orbitrap-Mass Spectrometry” may be changed as “… Ultra-High Performance Liquid Chromatography-High Resolution-Q-Orbitrap-Mass Spectrometry”
Lines 66-67, “…detect and quantify by High resolution accurate mass spectrometry small organic compounds.” Should be “…detect and quantify small organic compounds by high resolution accurate mass spectrometry.”
Lines 73-78, the first sentence is too complicated to be understood.
In table 1, there are several mistakes. The authors should check and revise them carefully. e.g., extra semicolon, obfuscated use of the symbols (“comma” is used in time and mass), “MS ions (ppm)” should be changed as “MS ions (m/z)”……
In table 1, the author claimed that the mass tolerance window was set to 5 ppm for the two modes, however, there are 11 compounds with the accuracy > 5 ppm.
Author Response
Responses to reviewer
Reviewer 2
This paper entitled “Phylogenetic studies and Metabolite Analysis of Sticta Species from Colombia and Chile by High Resolution UHPLC-Q-Orbitrap-Mass Spectrometry” by Albornoz et al., reported a work in the metabolomic analysis of 11 species of lichens extracts. The authors claimed that189 tentative analyses or features belong to five groups of compounds based on LC-MS data. In addition, the authors assessed the usefulness of chemical compounds in comparison to traditional morphological traits in the study of ancestor-descendant relationships in the genus. However, several questions should be addressed. The most important is the validation of those metabolites identification from lichens extracts; there is no method validation data (such as repeatability) for the detection of those compounds, so it is hard to evaluate the qualitative results.
Answer: We understand and appreciate the question asked by the reviewer. In this context, we have performed triplicate extractions and then, these lichen extracts mixed to obtain a single extract. This extract was analyzed by UHPLC/MS/MS. Although we should have three runs, we reduced these analyses to an extract and a single run.
The authors did not mention how they identify and annotate 189 tentative metabolites. ¿Can they be from a database or published references? Authors should provide evidence to show the credibility and accuracy for the identification of those endogenous metabolites. Some metabolite standards should be available to confirm their identification.
Answer: We thank the reviewer for his contributions to improve the manuscript. We agree with you, but we must take advantage with this dominant analytical technique in lichen chemistry. MS provide the elemental composition, isotope ratio, accurate m/z, and fragmentation patterns to allow a confirmation of a tentative structural determination of a molecule based on both published references and CFM-ID. These could establish a standart for the identification of compounds across untargeted metabolomic studies. However, the real structural determination come from the use of chemical standarts as the reviewer referred. Our research group use some standards to identified by spiking experiments as you can see at the bottom of the Table 1. In this study was used usnic acid, lobaric acid, galbinic acid, and fumarprotocetraric acid as published from us previously (sorry for the inconvenience caused for not including it).
The addition of the assessment of the usefulness of chemical compounds in comparison to traditional morphological traits is nice (although the compounds tree could not recover most evolutionary relations, authors claimed it can be used to distinguish the geographic clusters of collected samples.), but the quantitative analysis of those compounds has not been made. The authors carried out a phylogenetic study based on the presence/absence of compounds, however, are there any differences between biologic replicates collected from the same ecosystems and environmental conditions in South America?
Answer: Dear Reviewer, we appreciate this question. Few lichen studies have focused on contrasting the metabolites present in different biological samples taken from particular habitats, ecosystems, or regions. So, we are not sure how variable the presence of metabolites could be. Studies in plant species, using secondary metabolites, found geographic clusters of populations, which suggest a process of local adaptation (Pais et al. 2018). Given the close phylogenetic relationship among the samples (they represent the same genus), it is not unexpected that they could potentially share a broad kind of metabolites. However, the geographic clusters found in this study probably imply an emerging local adaptation process. Despite that all the samples were collected from the montain ecosystem, the different visited localities seem to show some particular conditions. Altitude, temperature, relative humidity, and wind speed could drive the establishment of potentially different communities of consumers, parasites, or other kinks of biological pressures.
Further, regarding the LC-MS data analysis, did the authors check and exclude the noise signals from the background? Did the authors check and eliminate the interferences from the in-source fragmentation (e.g., Peak 1 & Peak 4, Peak 23 & Peak 24, and Peak 36 & Peak 37)? Why did the authors only profile metabolites using UHPLC-MS in negative mode?
Answer: We appreciate the reviewer's question which allows us to make clarifications. Of course, we excluded the background and were checked out each peak to be included on the Table 1. A lot of unknown compounds were included since based on their elemental compositions is expected to be a lichen metabolites.
Answer:
Lines 3-4, “… High Resolution UHPLC-Q-Orbitrap-Mass Spectrometry” may be changed as “… Ultra-High Performance Liquid Chromatography-High Resolution-Q-Orbitrap-Mass Spectrometry”
Answer: Dear reviewer, we appreciate the suggestions made for the improvement of the manuscript, the changes have been made in the manuscript.
Lines 66-67, “…detect and quantify by High resolution accurate mass spectrometry small organic compounds.” Should be “…detect and quantify small organic compounds by high resolution accurate mass spectrometry.”
Answer: Dear reviewer, we appreciate the suggestions made for the improvement of the manuscript, the changes have been made in the manuscript.
Lines 73-78, the first sentence is too complicated to be understood.
Answer: Dear reviewer, we appreciate the suggestions made for the improvement of the manuscript, the changes have been made in the manuscript.
Answer:
In table 1, there are several mistakes. The authors should check and revise them carefully. e.g., extra semicolon, obfuscated use of the symbols (“comma” is used in time and mass), “MS ions (ppm)” should be changed as “MS ions (m/z)”……
Answer: Dear reviewer, we appreciate the suggestions made for the improvement of the manuscript, the changes have been made in the manuscript.
In table 1, the author claimed that the mass tolerance window was set to 5 ppm for the two modes, however, there are 11 compounds with the accuracy > 5 ppm.
Answer: Dear reviewer thank you for your nice corrections, it was added the phrase for the majority of compounds

Round 2
Reviewer 2 Report
Reviewer Report for the Authors
The work presented by Albornoz et al. shows the metabolite analysis of Sticta species from Colombia and Chile. The authors have addressed the reviewer’s comments and improved the quality of their manuscript. However, I have a few minor suggestions to improve the manuscript.
(1) In the Materials and Methods, the authors could clearly describe the sample preparation indicating that the LC-MS data came from a single run of a mixed extract.
(2) In addition, in Line 204, the footnote should be English.
Author Response
Responses to reviewer
Reviewer 2
The work presented by Albornoz et al. shows the metabolite analysis of Sticta species from Colombia and Chile. The authors have addressed the reviewer’s comments and improved the quality of their manuscript. However, I have a few minor suggestions to improve the manuscript.
(1) In the Materials and Methods, the authors could clearly describe the sample preparation indicating that the LC-MS data came from a single run of a mixed extract.
Answer: Dear reviewer, we are aware of your concerns, however, we would like to clarify that the LC-MS data are NOT from a single run of a mixed extract. Each lichen was extracted separately. We have clearly stated in the manuscript: The lichen extracts were then processed individually for the HPLC MS analyses (redissolved in methanol at a concentration of 1 mg/mL for the analyses), a sentence that has been incorporated in the paper.
(2) In addition, in Line 204, the footnote should be English.
Answer: We thank him for his contributions to the manuscript. We apologize for this error and have corrected it in the manuscript.
